# Glycosyl Formates: Glycosylations with Neighboring-Group Participation

**DOI:** 10.3390/molecules27196244

**Published:** 2022-09-22

**Authors:** Liang Yang, Christian Marcus Pedersen

**Affiliations:** Department of Chemistry, University of Copenhagen, Universitetsparken 5, 2100 Copenhagen Ø, Denmark

**Keywords:** glycosylation, catalysis, selectivity, formates, neighboring-group participation, atom economy

## Abstract

Protected 2-O-benzyolated glycosyl formates were synthesized in one-step from the corresponding orthoester using formic acid as the sole reagent. Glucopyranosyl, mannopyranosyl and galactopyranosyl donors were synthesized and their glycosylation properties studied using model glycosyl acceptors of varied steric bulk and reactivity. Bismuth triflate was the preferred catalyst and KPF_6_ was used as an additive. The 1,2-trans-selectivities resulting from neighboring-group participation were excellent and the glycosylations were generally high-yielding.

## 1. Introduction

Atom-economic glycosylation methods are receiving increasing interest as a consequence of the focus on green and sustainable chemistry. Early examples of simple catalytic glycosylations [1] in oligosaccharide synthesis came from the Kochetkov group, who introduced glycosylations using orthoesters [2]. Glycosyl fluorides [3] and trichloroacetimidates (TCAs) [4] appeared as glycosylation reagents in the early nineteen eighties and have since then been among the most popular glycosyl donors for catalytic glycosylations. ^1^
*N*-Phenyl trifluoroacetimidates have also received increasing attention as a glycosyl donor in recent years [5]. Although the abovementioned glycosyl donors are all effectively activated by simple Brønsted or Lewis acid catalysis, there has been a recent trend towards developing increasingly complex glycosylation methods involving expensive metal catalysts, and, often, complex leaving groups. The gold-catalyzed activation of alkyne derivatives has been a “hot topic” and has been used successfully in natural product synthesis [6,7]. Many of the newly developed methods are both elegant and useful [8], but most are restricted to small-scale synthesis due to the high cost of the specialized substrates and catalysts [9]. The use of highly specialized and sensitive reagents is to be avoided and heavy-metal promoters are environmentally concerning upon upscaling. Furthermore, large leaving groups generate more waste and, hence, a better atom economy is desirable. None of the methods mentioned above fulfill the demands for a scalable, cheap and environmentally friendly glycosylation method. Glycosyl esters are an interesting group of glycosyl donors, which are commonly used as anomeric protective groups. When used as an anomeric leaving group, relative harsh conditions are needed, which limits their use in oligosaccharide synthesis [1,10]. Methods using a less trivial remote activation have been developed to increase leaving group ability, but these methods require expensive reagents, such as Tf_2_O, in stoichiometric amounts [11]. Recently, several groups have reported glycosylation methods using ester leaving groups, catalyzed by metal triflates, but the scope is still limited [1,12,13,14,15]. Recently, we introduced glycosyl formates as glycosyl donors [16]. In addition to their simple synthesis using formic acid, this new donor type could be catalytically activated, and we have, furthermore, demonstrated that, under glycosylation conditions, the formic acid produced could be transformed into CO_2_ and hydrogen gas making the glycosylation “traceless”. In this preliminary study, it was found that Bi(OTf)_3_ was an excellent catalyst for activating the formyl group. Bi(OTf)_3_ has also been demonstrated to activate glycosyl halides [13,17] and the Ferrier rearrangement [18]. Working with the synthesis of glycosyl formates, it was surprising to find that this type of functional group had been only poorly investigated in glycosides. Hough and Lewis isolated 1-*O*-formoyl-β-d-glucopyranose tetraacetate as an undesired by-product, which represents the first description of a glycosyl formate [19]. Later, Vodcadlo and co-workers used formates as a precursor for the enol ether, which can be obtained via a Wittig-type reaction [20]. Frauenrath and coworkers prepared 1-*O*-vinyl glycosides from the corresponding formates using Tebbe’s reagent [21]. In this paper, our progress in using glycosyl formates as simple glycosyl donors is reported. The focus is on the synthesis of this new class of glycosyl donors and on their glycosylation properties, where neighboring-group participation is used for anomeric stereo control.

## 2. Results

With the goal of simplifying the glycosylation reaction and making it more atom-economic, we previously developed a new efficient synthesis of glycosyl formates, which avoids the use of reagents and catalysts. The hemiacetal of the donor was simply treated with formic acid giving the new donor and recovered starting material (Figure 1) [16]. The simplicity of the method contrasts with the previous syntheses of glycosyl formates, where glycosyl halides [19,20,21] thioglycosides [21] or coupling reactions (EDC) [21] were required. All these procedures produce at least a stoichiometric amount of waste.

It was observed that the glycosyl formate synthesis mainly achieved the α-product (α:β = 9:1), which could suggest thermodynamic control. Additionally, full conversion of the hemiacetal was not achievable, which is consistent with the establishment of an equilibrium between the products and the substrate.

When synthesizing 2-O-benzoyl-3,4,6-tri-*O*-benzyl-β-glucopyranosyl formate **D1** from the 1,2-orthoester **1**, it was found that the conditions using neat formic acid gave rise to partly debenzoylation and, therefore, it was decided to use diluted solutions of formic acid in CH_2_Cl_2_, which resulted in a very satisfying 89% isolated yield (Figure 2) [16]. The problems with incomplete formylations appear to be related to using the hemiacetal as the substrate, which liberated equivalent amounts of water upon esterification. Obviously, the use of orthoesters as substrates is desirable as these are readily available and the formed 2-benzoyl-group would be expected to impose neighboring-group participation, and even anchimeric assistance, in the otherwise relatively slow glycosylation reaction using formates [22,23]. Perbenzylated 1,2-orthoesters with *galacto-* and *manno*- stereochemistry (Appendix A: **S9** and **S13**) were therefore synthesized using literature protocols (see Appendix A for details). Treating these orthoesters with diluted formic acid resulted in a clean transformation into the corresponding glycosyl formates. Isolated yields of almost 90% with high anomeric selectivity clearly demonstrated the feasibility of this protocol (Figure 1; see Appendix A for details).

A common side-reaction, when using ester leaving groups in glycosylations, is hemiacetal formation from the nucleophilic attack on the ester carbonyl carbon, i.e., a transesterfication. This side-reaction, quantified by the “GT” parameter, has previously been shown to be dependent on additives, catalyst loading, and the nucleophile used [16]. In this investigation, our hypothesis was that having neighboring-group participation would reduce this reaction pathway via the formation of a dioxolenium ion intermediate faster than the transesterification reaction takes place.

With access to 2-O-benzoylated glycosyl formates (**D1** to **D3**) with different stereochemistry, evaluation of their glycosylation properties could begin. The glucosyl donor **D1** has previously been used [16] and, hence, served as a reference for the new donors. The previous results are included in the table for completeness.

Due to the relative long reaction times with armed glycosyl donors [24], disarmed glycosyl donors carrying more acyl groups were not investigated in depth and an initial screening confirmed the low reactivity.

Four model glycosyl acceptors, with different steric bulk and nucleophilicities, were chosen for this study (Figure 1). Cyclohexanol **A1** was used as a secondary achiral alcohol with medium steric bulk. 1-Adamantanol **A2** is a sterically more demanding glycosyl acceptor although still nucleophilic despite being a tertiary alcohol [25]. A 6-OH and a 4-OH glucosyl acceptor (**A3** and **A4**) were also included as more relevant and challenging nucleophiles, with different reactivities. Glycosylation using **D1** was carried out under standard conditions, i.e., THF, 25 mol%, Bi(OTf)_3_ and KPF_6_ as an additive. To our surprise, the least reactive nucleophile **A4** was the best performing, giving an 81% isolated yield of the β-anomer **G4**. Both **A2** and **A3** also gave good, isolated yields around 70% of the β-anomers (**G2** and **G3,** respectively). Glycosylations with donor **A1** improved the β-selectivity and the yield remained similar to the reactions with a 2-OBn group (Table 1, entry 1–4, and previous results with the perbenzylated glucosyl formate [16]). Turning to the galactosyl donor **D2**, acceptors **A1**–**A3** all gave similar yields of 70% and, exclusively, the β-galactosides **G5**–**G7**, consistent with the results from the glucosylations (Table 1, entry 5–7). However, when using the more challenging 4-OH acceptor **A4,** the yield was reduced and only 38% of the β-galactoside **G8** could be isolated (Table 1, entry 8). Attempts to increase this yield significantly were not successful and, hence, the current methods have some limitations when combining the galactosyl donor with bulky, less-reactive glycosyl acceptors. The main side-product was found to be the hemiacetal, suggesting reaction at the formate carbonyl by the nucleophile and, hence, a more dominant transesterification pathway. The axial 4-O-Bn in the galactosyl formate is assumed to hinder the access to the β-side and, hence, to disfavor galactoside formation, when a bulky acceptor is used. This pushes the reaction towards transesterification, as illustrated with the blue arrows (Figure 3). Mannosylations using **D3** (entry 9–12) followed the trend observed with the glucosylation and gave good isolated yields of **G9**–**12**, all with excellent 1,2-trans selectivity. The more hindered nucleophiles again performed slightly better, as was observed with the glucosyl donor **D1**. This can be explained by a slower reaction, which predominantly passes through a dioxolenium ion intermediate, and, hence, diminishes the reaction of the formate carbonyl.

Motivated by the generally good, isolated yields and excellent stereoselectivity, the study was expanded to include another common class of protective groups known to take part in neighboring-group participation, i.e., carbamates. The *N*-Alloc-protected glucosyl donor **D4** was, therefore, prepared and studied using the standard conditions. As this donor cannot be synthesized directly from an orthoester, a new approach had to be developed. As described in our recent paper, hemiacetals can be used as substrates for the introduction of the formyl group. The corresponding hemiacetal was therefore prepared, inspired by literature procedures [26,27]. Formylation using formic acid could, however, not be used on this substrate, and mixed anhydrides were, therefore, investigated. Acetic formic anhydride was found to be the best choice, giving the donor **D4** in 87% (brsm) yield. In contrast to the 2-O-benzoyl-protected glycosyl donors, no glycosylation products were observed, and the main product obtained was, disappointingly, the hemiacetal. This suggests that the alloc group does not effectively participate in the reaction by neighboring-group participation and that the nucleophile, therefore, primarily reacts at the formyl group. Together with similar results using per benzoylated glycosyl formates (not shown), this suggests that, when the glycosyl donor is becoming too disarmed, i.e., less reactive, the reaction pathway leading to the hemiacetals becomes dominant. Changing the acceptor did not change this outcome of the glycosylations and, hence, this method has a limitation when it comes to the functionality on position 2.

## 3. Conclusions

In conclusion, we have developed a general method for synthesizing various glycosyl formates with a 2-O-benzoyl group from the corresponding orthoester. The reaction proceeds under mild conditions with formic acid as the sole reagent. The 2-*O*-benzoyl group installed secures neighboring-group participation and, hence, 1,2-trans-selective glycosylations. The new donors were found to be storable and easy to handle. Glycosylations were performed using Bi(OTf)_3_ in catalytic amounts and with KPF_6_ as an additive. The glycosides were generally isolated in high yields and with high 1,2-trans-selectivity.

## Data Availability

NMR spectra and detiled experimental descriptions can be found in the supporting information.

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
