# Peer review of "Glycosyl Formates: Glycosylations with Neighboring-Group Participation"

_molecules, 2022, doi:10.3390/molecules27196244_

Round 1
Reviewer 1 Report
The authors successfully described one-step synthesis of glycosyl formates and glycosylation properties of these donors in presence of NGP. The synthetic methodology and experimental steps are well described, and purity of the compounds are satisfactory. I recommend accepting this manuscript after the minor revisions as suggested below.
1. Add melting point of the solid compounds.
2. For 500 MHz NMR the 13C NMR are recorded at 125 MHz, please check that.
3. Add structure of the products (disaccharides) with numbers in figure 1.
4. Add compound numbers of the disaccharides in the Table 1.
5. Structure of S20 (Formed by reaction between D2 and A4) in supporting information file has drawn as a but describes as b. please check and correct.
6. Please include the following references for recent review, application of gold in activation of alkynes, Bi(OTf)3 in carbohydrate chemistry.
a) Front. Mol. Biosci., 9, 2022, 896187. doi: 10.3389/fmolb.2022.896187
b) Chem. Asian J. 14, 4651–4658. doi:10.1002/asia.201900888
c) J. Am. Chem. Soc. 141 (21), 8509–8515. doi:10.1021/jacs.9b01862
d) Org. Biomol. Chem., 2021, 19, 3220, doi: 10.1039/d1ob00093d
e) Org. Lett. 2022, 24, 575−580 doi: org/10.1021/acs.orglett.1c04008
f) Chem. Soc. Rev. 2011, 40, 4649, DOI: 10.1039/c0cs00206b (References therein)
g) Molecules, 2005, 10, 884-892
Author Response
Reply: Thanks for reviewing our paper. We have done our best to improve our manuscript. Below is a point to point reply to the critics/comments raised by the reviewer:
- The new compounds isolated as solids have all been obtained by column chromatography and hence not recrystallized. Melting points have therefore not been obtained.
- The 13C NMR has been recorded at 126 MHz (rounded value)
- A generalized scheme has been added to Table 1.
- A general glycosylation scheme has been included in connection with table 1 and product number have been added to the table (G1-12). The product numbers have been included in the text as well. The SI has been updated with the new numbering.
- The structure of S20 was wrong and has been corrected – thanks!
- We have went through the references suggested and included the ones relevant for this work. The text has been changed accordingly.
Reviewer 2 Report
Dr. Christian Marcus Pedersen and coworkers described about the "glycosyl formates and neighbouring group participation". This manuscript can be accepted tor publication, only after addressing the following issues.
1. Title should not be a fragmented words. It should reflect the content clearly.
2. Abstract section: The sentence does not implies anything "Donors with gluco-, manno- and galacto- stereochemistry have been prepared in high yields and their glycosylation properties studied." stereochemistry of the sugar can not be prepared. Need major modifications.
3. Neighbouring group participation (NGP) is not clearly demonstrated in this manuscript. Kinetic studies would give more information about the NGP effect.
4. Author didn't show the calculation related to the formation of alpha anomeric product in the case of formate ester synthesis.
5. This manuscript does not contain the uniformity. Example some places, the compound numbers are numberic (1) and some places with alphabets (D1) and some other places there's no mention about the compound (please refer Scheme 1).
6. Conclusion section is very weak and it does not talk about the actual work.
7. Authors used "we" in several places. It should have been avoided.
8. Interpretation of each peak in NMR give more information to the reader.
Author Response
Reply: Thanks for reviewing our paper. We have followed the valuable points raised by the reviewer and improved our manuscripts accordingly. Below is a point to point reply to the critics/comments raised by the reviewer:
- Thanks for the suggestion to make the title more clear. The title has now been changes to “Glycosyl Formates: Glycosylations with Neighboring Group Participation”
- The abstract has now ben rewritten: Abstract: Protected 2-O-Benzyolated glycosyl formates have been synthesized in one-step from the corresponding orthoester using formic acid as the sole reagent. Glucopyranosyl, mannopyranosyl and galactopyranosyl donors were synthesized and their glycosylation properties studied using model glycosyl acceptor of various steric bulk and reactivity. Bismuth triflate was the preferred catalyst and KPF6 was used as an additive. The 1,2-trans-selectivities resulting from neighboring group participation were excellent and the glycosylations were generally high yielding.
- The neighboring group participation is evident from the high 1,2-trans selectivity, which is an established and generally accepted method in glycosylation chemistry. Whether the reactions are becoming faster (kinetics) we have not studied in detail and hence we cannot specify whether anchimeric assistance is taking place.
- When synthesizing the glycosyl formates only the 1,2-trans products were isolated as stated in the SI (compound names).
- Numbers for products have now been included with a letter end a number (G1, G2 etc.) Other structure numbers does not related to the glycosylation reactions.
- The conclusion has been rewritten to: “formates, having a 2-O-benzoyl group, from the corresponding orthoester. The reaction proceed under mild conditions with formic acid as the sole reagent. The 2-O-benzoyl group installed secures neighboring group participation and hence 1,2-trans-selective glycosylations. The new donors were found to be storable and easy to handle. Glycosyla-tions were performed using Bi(OTf)3 in catalytic amounts and with KPF6 as an additive. The glycosides were generally isolated in high yields and with high 1,2-trans selectivity.”
- The numbers of “we” has been reduced and the text rewritten accordingly.
- I fully agree! And most NMR signals of new compounds have therefore been assigned.